

# Movements and dispersal of brown trout (*Salmo trutta* Linnaeus, 1758) in Mediterranean streams: influence of habitat and biotic factors

Enric Aparicio[1,*], Rafel Rocaspana[2], Adolfo de Sostoa[3], Antoni Palau-Ibars[4] and Carles Alcaraz[5,*]

[1] GRECO, Institute of Aquatic Ecology, University of Girona, Girona, Catalonia, Spain
[2] Gesna Estudis Ambientals, Linyola, Catalonia, Spain
[3] Department of Evolutionary Biology, Ecology and Environmental Sciences, University of Barcelona, Barcelona, Catalonia, Spain
[4] Department of Environment and Soil Sciences, University of Lleida, Lleida, Catalonia, Spain
[5] IRTA Marine and Continental Waters, Sant Carles de la Ràpita, Catalonia, Spain
[*] These authors contributed equally to this work.

Corresponding author
Enric Aparicio,
enric.aparicio@gmail.com

## ABSTRACT

Dispersal is a critical determinant of animal distribution and population dynamics, and is essential information for management planning. We studied the movement patterns and the influence of habitat and biotic factors on Mediterranean brown trout (*Salmo trutta*) by mark-recapture methods in three headwater streams of the Ebro Basin (NE Iberian Peninsula). Fish were sampled by electrofishing on five occasions over 18–24 months and movements of over 3,000 individually tagged trout (age 1+ onwards) were recorded. Most of the tagged fish exhibited limited movement and were recaptured within 100 m from the initial capture section. Small seasonal differences in the movement pattern were observed, but in two of the streams, displacement distances increased prior the spawning period in autumn. The frequency distributions of dispersal distances were highly leptokurtic and skewed to the right and fitted well to a two-group exponential model, thus trout populations were composed of mobile and stationary individuals, the latter being the predominant component in the populations (71.1–87.5% of individuals). The mean dispersal distances, for fish captured at least in three sampling events, ranged 20.7–45.4 m for the stationary group and 229.4–540.5 m for the mobile group. Moving brown trout were larger than non-moving individuals and exhibited higher growth rates in two of the streams. Habitat features were not consistently linked to movement rates, but there were some interaction effects between stream and habitat characteristics such as depth, cover and water velocity.

## INTRODUCTION

Dispersal, defined as the movement of individuals between locations, is a key process that allows fish to occupy the most suitable habitat for survival, growth and breeding by adapting to both temporal and spatial changes in environmental or biotic conditions (*Railsback et*

*al., 1999*; *Lucas & Baras, 2001*). Fish movement can be highly variable between individuals, species and streams (*Rodríguez, 2002*). Most research on fish dispersal in streams shows heterogeneous populations comprising both sedentary and mobile individuals (*Rodríguez, 2002*; *Rasmussen & Belk, 2017*). The movement distribution (i.e., distance moved *vs.* probability of occurrence) is then best predicted by leptokurtic dispersal models which are characterized by high peaks associated with the sedentary fish, and longer tails associated with a lower proportion of mobile fish (*Skalski & Gilliam, 2000*; *Rodríguez, 2002*). This modelling framework has been applied to describe the general movement patterns of fish and a predictive tool was recently developed based on the positive correlation of movement distances with four variables: fish length, aspect ratio of the caudal fin, stream size, and duration of the study (*Radinger & Wolter, 2014*). At a local scale, both abiotic and biotic factors affect the degree of movement, variability and dispersal rates. For instance, some studies have reported an increase in exploratory behaviour (i.e., higher proportions of mobile fish) with decreasing habitat heterogeneity (*Albanese, Angermeier & Dorai-Raj, 2004*; *Heggenes et al., 2007*); the presence of fish cover (e.g., woody debris and boulders) has been related to a reduction in fish movement (*Aparicio & Sostoa, 1999*; *Harvey, Nakamoto & White, 1999*); and population density has been positively linked to fish movement rates (*Hesthagen, 1988*).

The mobility and dispersal patterns of brown trout *Salmo trutta* Linnaeus, 1758, have been reported to be largely variable. Some populations are mainly sedentary (*Northcote, 1992*; *Burrell et al., 2000*; *Knouft & Spotila, 2002*), while others are dominated by individuals that move extensively (*Clapp, Clark Jr & Diana, 1990*; *Meyers, Thuemler & Kornely, 1992*; *Ovidio et al., 1998*). The movement patterns of brown trout have been widely studied within its native range in central and northern Europe, as well as in its introduced range in North America (see *Rodríguez, 2002*, and references therein), but the information on brown trout movements in rivers flowing to the Mediterranean Sea, in its southern native range, is scarce. There is only a study in the Rhône River focussed on the movements of age 0+ individuals (*Vatland & Caudron, 2015*). Brown trout populations in the Mediterranean Basin are highly genetically differentiated from central and northern Europe populations and are considered as distinct lineages (*Bernatchez, 2001*; *Cortey, Pla & García-Marín, 2004*). These populations have experienced a marked decline in the last decades due to stream habitat degradation (*Benejam et al., 2016*), overfishing (*Almodóvar & Nicola, 2004*), and introgressive hybridization with hatchery stocks (*Aparicio et al., 2005*). Climate change scenarios for the Mediterranean predict additional reductions in brown trout distributional range, due to rising water temperatures (*Almodóvar et al., 2012*). Consequently, there is an urgent need to develop sound conservation and management strategies for preserving these populations. With increasing river fragmentation, quantifying the scale and magnitude of Mediterranean brown trout dispersal patterns and determining the factors that drive these, are necessary to optimize management planning.

The overall purpose of this study was to examine the movement patterns of brown trout populations from three separate streams of a Mediterranean catchment of the Iberian Peninsula and the influence of biotic and abiotic factors. The three study streams are small, have a high gradient, and considerable habitat heterogeneity. When all habitats required to

complete the cycle of a fish's life-history are in spatial proximity, lower movement distances are expected (*Albanese, Angermeier & Dorai-Raj, 2004*). Also, structural habitat complexity increases protection from predators, refuge from disturbances (e.g., flooding) and prey availability, reducing territory size and fish mobility (*Heggenes et al., 2007*; *Závorka et al., 2015*). The objectives of this study were to employ mark-recapture approaches over a period of 18–24 months to determine (1) the degree of mobility (i.e., proportion of stationary and mobile individuals) and extent of displacement distances, (2) seasonal variations in the pattern of movements, and (3) the effect of habitat characteristics and biotic factors on trout movement.

## MATERIAL AND METHODS

### Study area

The marking and recapturing of individuals were conducted in three streams (Noguera Pallaresa, Flamisell and Noguera Vallferrera) of the Segre River basin (Catalonia, NE Iberian Peninsula) (Fig. 1). The Segre is the largest catchment in the Southern Pyrenees (265 km long, 22,580 km$^2$ of basin area and about 100 m$^3$ s$^{-1}$ of average water flow), and is the main tributary of the Ebro River, the river with the highest water flow in the Iberian Peninsula (annual mean is about 426 m$^3$ s$^{-1}$) discharging into the Mediterranean Sea (for more details see *Rovira, Alcaraz & Ibáñez, 2012*). A tributary of the Segre River is the Noguera Pallaresa River (154 km of length and 37.1 m$^3$ s$^{-1}$ of average flow), where three different reaches were selected (Fig. 1): the main stem (42°44′N, 0°58′E, hereafter NP) and two tributaries, the Flamisell (42°27′N, 0°59′E, FLM) and the Noguera Vallferrera (42°34′N, 1°19′E, NV). The three reaches had a mean stream width less than 10 m, showed a high gradient, and channel morphology consisted primarily of pool-run-riffle sequences under a forest canopy. Physical characteristics of study reaches are shown in Table 1. The hydrological regime is snow-fed, thus, the highest flows generally occur in spring after snowmelt (Ebro Water Authority; http://www.chebro.es/). Brown trout was the only fish species present and populations belong to the Mediterranean lineage (*Aparicio et al., 2005*). During the study, sport fishing was closed in the study area.

### Sampling design

Study reaches were continuous and ranged from 2,400 m long in the NP, 1,500 in the FLM to 1,200 m long in the NV. The reaches were selected on the basis of being wadeable and the absence of barriers to fish movement or significant tributaries within the study boundaries. Each reach was divided into 25–40 m-long sections (mean length: 30.6 m ± 4.3 SD), the boundaries of which usually coincided with habitat discontinuities in the channel morphology (pool-run-riffle sequence). Brown trout were marked in October 1992 in NP and in March 1993 in NV and FLM. Four subsequent mark-recapture sampling events were conducted in each reach: NP was sampled in 1993 (May and October) and 1994 (July and October), and both NV and FLM reaches were sampled in 1993 (July and October) and 1994 (July and October). Autumn surveys (October) were performed just prior or at the beginning of the spawning season, which in the study area occurs from the end of

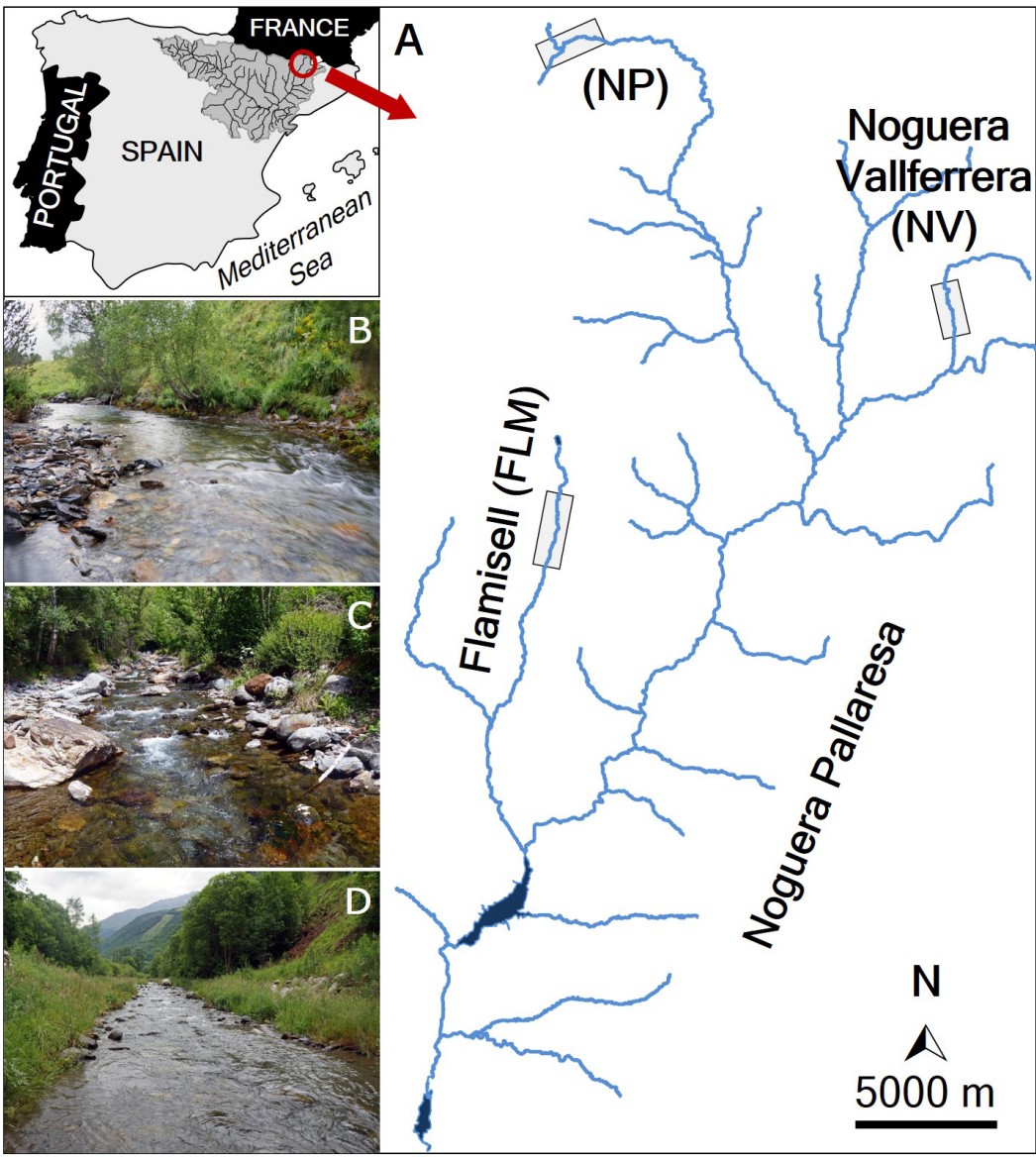

**Figure 1** **Map of the study area.** (A) Location of the study streams: Noguera Pallaresa (NP), Flamisell (FLM) and Noguera Vallferrera (NV). Location of the sampled stream reaches is highlighted with a rectangle. The photos show representative habitats in the streams sampled. (B) NP. (C) NV. (D) FLM. Photographs by Enric Aparicio.

October to early December. Therefore, movement data from October sampling included the possible fish displacements related to spawning. Sampling sections were isolated with block nets (mesh size 5 mm) and three-pass electrofishing removal was conducted in an upstream direction (500 V, 1.0 A pulsed DC). Captured fish were anaesthetised (MS-222 solution), measured for fork length (FL, mm), and weighed (g). Sex was determined externally for mature individuals in autumn (i.e., October) surveys by the production of

**Table 1 Stream features during the sampling period.** FLM: Flamisell, NP: Noguera Pallaresa, and NV: Noguera Vallferrera. Mean ± standard deviation, when necessary, is shown.

| Variable | Stream | | |
|---|---|---|---|
| | **FLM** | **NP** | **NV** |
| Length (m) | 1,500 | 2,400 | 1,200 |
| Mean elevation (m) | 1,250 | 1,780 | 1,175 |
| Mean slope (m km$^{-1}$) | 53.3 | 20.8 | 62.5 |
| Mean base flow (m$^3$s$^{-1}$) | 1.52 | 0.95 | 1.29 |
| Temperature range (°C) | 4.2–15.3 | 0.1–15.1 | 2.5–15.4 |
| pH range | 7.3–8.9 | 7.4–8.8 | 6.7–8.1 |
| Conductivity range (μS cm$^{-1}$) | 77–171 | 46–187 | 20–95 |
| Mean stream width (m) | 4.97 ± 1.39 | 4.34 ± 2.36 | 6.80 ± 1.09 |
| Mean water depth (cm) | 9.94 ± 4.66 | 20.02 ± 6.46 | 17.14 ± 2.77 |
| Mean current velocity (m s$^{-1}$) | 0.66 ± 0.26 | 0.59 ± 0.27 | 0.70 ± 0.14 |
| **Mean bottom substrate (%)** | | | |
| Boulders (>256 mm) | 31.47 ± 13.82 | 30.84 ± 18.74 | 48.87 ± 12.67 |
| Cobble (65–256 mm) | 34.14 ± 11.89 | 34.68 ± 15.17 | 32.45 ± 10.95 |
| Gravel (2–65 mm) | 15.01 ± 6.17 | 14.89 ± 10.37 | 7.69 ± 3.04 |
| Sand (0.1–2 mm) | 10.60 ± 4.23 | 10.77 ± 10.16 | 3.74 ± 3.04 |
| Silt (<0.1 mm) | 3.45 ± 2.55 | 4.41 ± 4.54 | 2.04 ± 1.13 |
| Organic | 4.33 ± 3.04 | 4.13 ± 3.17 | 5.17 ± 3.01 |
| Mean fish density (Ind ha$^{-1}$) | 1,345.9 ± 1,322.2 | 1,901.4 ± 989.3 | 3,862.5 ± 1,559.0 |
| Mean fish biomass (Kg ha$^{-1}$) | 67.5 ± 59.8 | 107.8 ± 82.8 | 344.7 ± 182.7 |
| Mean fish cover (%) | 15.46 ± 8.81 | 10.18 ± 7.57 | 22.47 ± 6.52 |

milt or visible evidence of eggs beneath the body wall. The adipose fin was clipped on all captured fish, which constituted a permanent batch mark to distinguish recaptured individuals.

All fish larger than 120 mm FL (corresponding at least to age 1+ individuals, see Figs. S1–S3) were tagged with uniquely coded Visible Implant Tags (VI-tags; Northwest Marine Technology, Shaw Island, Washington, USA) in the adipose tissue posterior to eye. Fish smaller than 120 mm were not tagged, because the tissue available for placing the tag was too thin, thus leading to extremely low retention rates (*Niva, 1995*). After handling, fish were allowed to recover from anaesthesia and released into the mid-point of the capture section. In recapture surveys, each fish's VI-tag code was read, and those untagged fish larger than 120 mm FL (i.e., captured for the first time or having lost the VI-tag) were tagged following the above procedure.

Habitat variables were measured along transects perpendicular to the flow at 10-m intervals. Water depth and mean water velocity (at 0.6 times total water depth) were measured every 1m along transects, and averaged for each section. For each transect, the percent cover of each substrate category (Table 1) was visually estimated, averaged per section and an index of substrate coarseness was derived for each section following *Bain, Finn & Booke (1985)*. Fish cover was estimated as the percentage area of a section covered by woody debris, rocks or undercut banks.

Permission for electrofishing and capture of *S. trutta* individuals was approved by the competent authorities: Departament de Medi Ambient i Habitatge de la Generalitat de Catalunya (current Departament d'Agricultura, Ramaderia, Pesca, Alimentació i Medi Natural) (SF/602) of the regional authorities of Catalonia.

## Data analyses

Fish movement (i.e., distance covered) was measured from the midpoint of the recapture section to the midpoint of its previous capture section. A movement distance of 0 m was assigned to individuals captured and recaptured in the same section. Thus, the minimum detectable movement was the distance between two or more sections (i.e., larger than 25–40 m). Movement patterns were quantified using frequency distributions (two-tailed) of distances moved. Positive values were assigned to upstream movements and negative values to downstream movements. Mark-recapture studies are generally biased since long movements are less frequently measured than short distances (*Albanese, Angermeier & Gowan, 2003*). To assess this source of bias, all movement observations were weighted by determining the total possible movements sampled for each distance and weighting the under-sampled distances (*Porter & Dooley, 1993*; *Albanese, Angermeier & Gowan, 2003*). The movement distributions were not significantly different after adjustment (Kolmogorov–Smirnov test, $P > 0.40$ for all comparisons), suggesting a good study design (*Porter & Dooley, 1993*). Therefore, unweighted distributions were used for all analyses.

Trout were classified as moving (individuals leaving the home section between sampling events) and non-moving (sedentary individuals not leaving the home section) individuals. Differences in distance covered by moving fish among streams and sampling events were analysed with analysis of variance (two-way ANOVA). The kurtosis and skewness of the frequency distribution of distances moved by brown trout were obtained with the package *moments* (version 0.14) for the program R (*Komsta & Novometsky, 2015*), and were used as an indicator of individual level variation in movement behaviour (*Skalski & Gilliam, 2000*). Dispersal range was estimated as the distance between the two most distant sections in which an individual was recorded, and only fish captured at least in three sampling events (elapsed time between first and last capture was 7 to 24 months) were included in the analyses. The frequency distribution of dispersal range was fitted to a two-group exponential function (*Rodríguez, 2002*):

$$f(x) = p\frac{\lambda_s}{2}e^{-\lambda_s x} + (1-p)\frac{\lambda_m}{2}e^{-\lambda_m x}$$

where $x$ is the distance covered, $\lambda_s$ correspond to the inverse of mean dispersal distance for the stationary component, $\lambda_m$ correspond to the inverse of mean dispersal distance for the mobile component, $p$ is the proportion of stationary individuals, and thus, $1-p$ is the proportion of mobile individuals. Parameters $p$, $\lambda_s$ and $\lambda_m$ are estimated from movement data (see *Rodríguez, 2002*). Estimated movement parameters were then compared to the expected movement parameters for stream-resident brown trout obtained with the package *fishmove* (version 0.3–3) for the program R, which models dispersal in relation to species, fish length, aspect ratio of the caudal fin, stream size and duration of the study (*Radinger & Wolter, 2014*).

In order to examine the relationship between fish growth rate and movement, the specific growth rate (SGR, in % day$^{-1}$) for a given fish in the summer period was calculated as SGR = (ln(final fork length)—ln(initial fork length)) $\times 100$/(days between captures). Differences in the growth rate–fork length relationship between moving and non-moving individuals were tested with analysis of covariance (ANCOVA), for each stream. Software Pop/Pro 1.0 (*Kwak, 1992*) was used to estimate population size per section and capture probability per size class (i.e., trout smaller and larger than 120 mm) from the three-pass electrofishing data. Fish density (individuals ha$^{-1}$) and biomass (kg ha$^{-1}$) per section were estimated by dividing number and weight by the area sampled. Variations in fish density and biomass among streams were analysed with multiple analysis of variance (MANOVA). MANOVA is used when several dependent variables are measured on each sampling unit. MANOVA compares the mean vectors of $k$ groups; whereas equality of the mean vector implies that the $k$ means are equal for each variable, if two means differ for just one variable then we conclude that the mean vectors of the $k$ groups are different (*Sokal & Rohlf, 1995*). We used $\eta^2$ (eta squared) as a measure of effect size (i.e., importance of factors). Similarly to $r^2$, $\eta^2$ is the proportion of variation explained for a certain effect.

The effects of biotic and abiotic stream features of sampling sections on trout departure ratio (defined as the proportion of individuals leaving section $i$ from the total number of recaptures of trout initially marked in section $i$) were analysed with Generalized Linear Models (GLMs). We performed a global analysis including the three different streams, and then streams were analysed separately. In each case, an information-theoretic approach was used to find the best approximating models (*Burnham & Anderson, 2002*). GLMs were built including all possible combinations of environmental and biotic variables, excluding interactions, due to the large number of variables included. Two additional criteria were used to define the set of candidate models: those performing significantly better than the null model and with a variance inflation factor $\leq 5$, in order to avoid multicollinearity effects. The second order Akaike Information Criterion (AICc) was used to assess the degree of support for each candidate model. AICc was rescaled to obtain $\Delta$AICc values ($\Delta$AICc = AICc$_i$—minimum AICc), since models with $\Delta$AICc $\leq 2$ have the most substantial support. The relative plausibility of each candidate model was assessed by calculating Akaike's weights ($w_i$), which range from 0 to 1, and can be interpreted as the probability that a given model is the best model in the candidate set. Because no model was clearly the best one (i.e., $w_i \geq 0.9$) model-average regression coefficients were calculated using the relative importance of each independent variable (*Burnham & Anderson, 2002*). Model-averaged coefficients were compared with those from the full model to assess the impact of model selection bias on parameter estimates. All data analyses were performed with R software version 3.3.2.

# RESULTS

## Recapture rates and VI-tag retention

A total of 13,340 individuals were captured and fin-clipped during the study period, and 7,050 of them were larger than 120 mm FL and tagged (NP: 2201; FLM: 1181; NV: 3668).

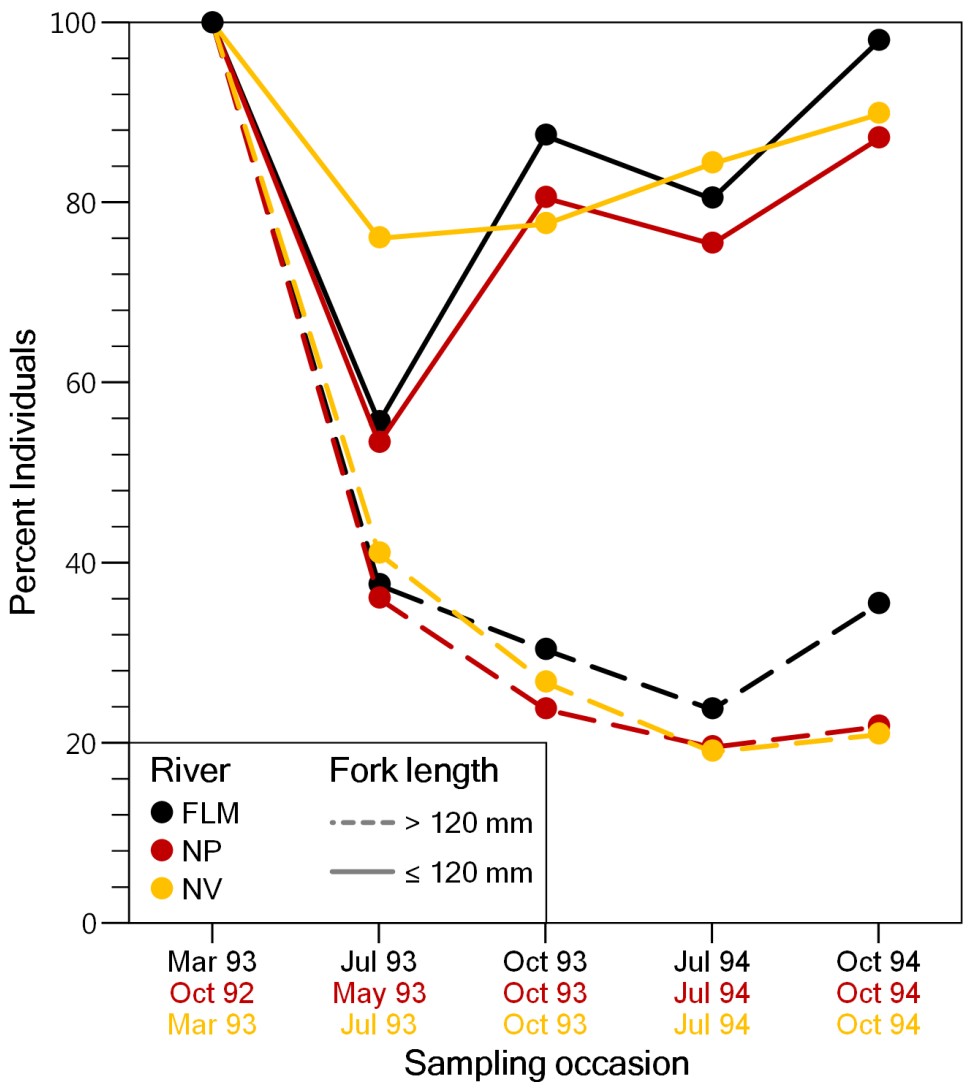

**Figure 2** **Evolution of the percentage of brown trout >120 mm fork length and ≤120 mm fork length captured for the first time (i.e., adipose fin not clipped), on each sampling event in the study streams.** FLM, Flamisell; NP, Noguera Pallaresa,; NV Noguera Vallferrera.

Mean FL of tagged fish was 165.1 mm ± 34.7 SD in the NP, 162.8 mm ± 38.1 SD in the FLM and 183.0 mm ± 42.1 SD in the NV. Capture probabilities estimated from electrofishing depletion surveys ranged from 0.40 to 0.88 ($\mu = 0.65$) for trout larger than 120 mm, and from 0.14 to 0.49 ($\mu = 0.38$) for individuals smaller than 120 mm. The proportion of the new trout individuals (i.e., fish not fin-clipped) larger than 120 mm FL captured for the first time was similar among streams and decreased sharply until the end of the study, with a maximum reduction of about 60% between the first and the second sampling (Fig. 2). The decline in the proportion of new individuals smaller than 120 mm FL captured in the second survey was less pronounced (Fig. 2), probably because of recruitment and reduced capturability. Despite the high recapture rates, the recovery percentage of VI-tag (i.e., fish

**Table 2** **Parameter estimates from the two-group exponential model to calculate the proportion of stationary ($p$) and mobile ($1 − p$) individuals, and mean dispersal range ($1/\lambda$) of brown trout population.** FLM: Flamisell, NP: Noguera Pallaresa, and NV: Noguera Vallferrera. $\lambda$ ($m^{-1}$) is the inverse of the mean displacement range, lower values of $\lambda$ correspond to more mobile individuals.

| Stream | Population component | Proportion (%) | Parameter estimate | Mean dispersal range (m) |
|--------|---------------------|----------------|--------------------|--------------------------|
| FLM | Stationary | 87.5 | $\lambda_s = 0.02201$ | 45.4 |
| | Mobile | 12.5 | $\lambda_m = 0.00185$ | 540.5 |
| NP | Stationary | 85.6 | $\lambda_s = 0.04184$ | 23.9 |
| | Mobile | 14.4 | $\lambda_m = 0.00269$ | 371.7 |
| NV | Stationary | 71.1 | $\lambda_s = 0.04822$ | 20.7 |
| | Mobile | 28.9 | $\lambda_m = 0.00436$ | 229.4 |

larger than 120 mm) was relatively low due to tag loss. Overall, mean VI-tag retention rate was 45.0% but varied in relation to fish size. Trout smaller than 180 mm FL at tagging showed mean retention rates of 32.4% whereas the rate was 74.9% for trout larger than 240 mm FL.

## Movements and dispersal patterns

Overall, movements of brown trout between consecutive sampling events were relatively short since the 53.4% (ranging between 48.1 and 54.4%) of the total recaptures ($N = 3,073$) were in the same stream section of previous capture. When fish moving out from home section were analysed separately, most of the displacements were between contiguous sections, and long-range movements were infrequent; for instance, only the 2.18% of the movements were over 400 m in NV, being 2.37% in NP and 3.35% in FLM. Distance covered by fish moving out from home section was not significantly related to fork length (ANCOVA, $P > 0.29$ in all rivers) or length $\times$ season interaction ($P > 0.15$), so an ANOVA was used. Distance covered by fish showed significant temporal variations in both the NP (ANOVA, $F_{3,426} = 4.49$, $P = 0.004$, $\eta^2 = 0.24$) and NV ($F_{3,795} = 9.45$, $P < 0.0001$, $\eta^2 = 0.26$) streams, with trout covering larger distances in autumn (Fig. 3), but seasonal differences were not significant in the FLM stream ($F_{3,198} = 1.08$, $P = 0.36$).

In the three streams the distance-frequency distributions were highly leptokurtic and skewed to the right, the Kurtosis values were 28.9 for FLM, 19.6 for NV and 144.6 for NP. The two-group exponential model provided a good fit to the frequency distributions of dispersal range (Fig. 4). Based on the estimates of $p$, the proportion of stationary individuals in the three streams ranged from 71.1 to 87.5% and, thus, the percentage of mobile individuals varied from 12.5 to 28.9% (Table 2). The mean dispersal range, calculated as $1/\lambda$ (see 'Methods'), ranged from 20.7 to 45.4 m for the stationary group and from 229.4 to 540.5 m for the mobile group (Table 2). These dispersal ranges were similar to that expected by the general model for stream resident brown trout estimated with *fishmove* package, which calculates a mean movement distance of 25.1 m for the stationary group and 428.3 m for the mobile group.

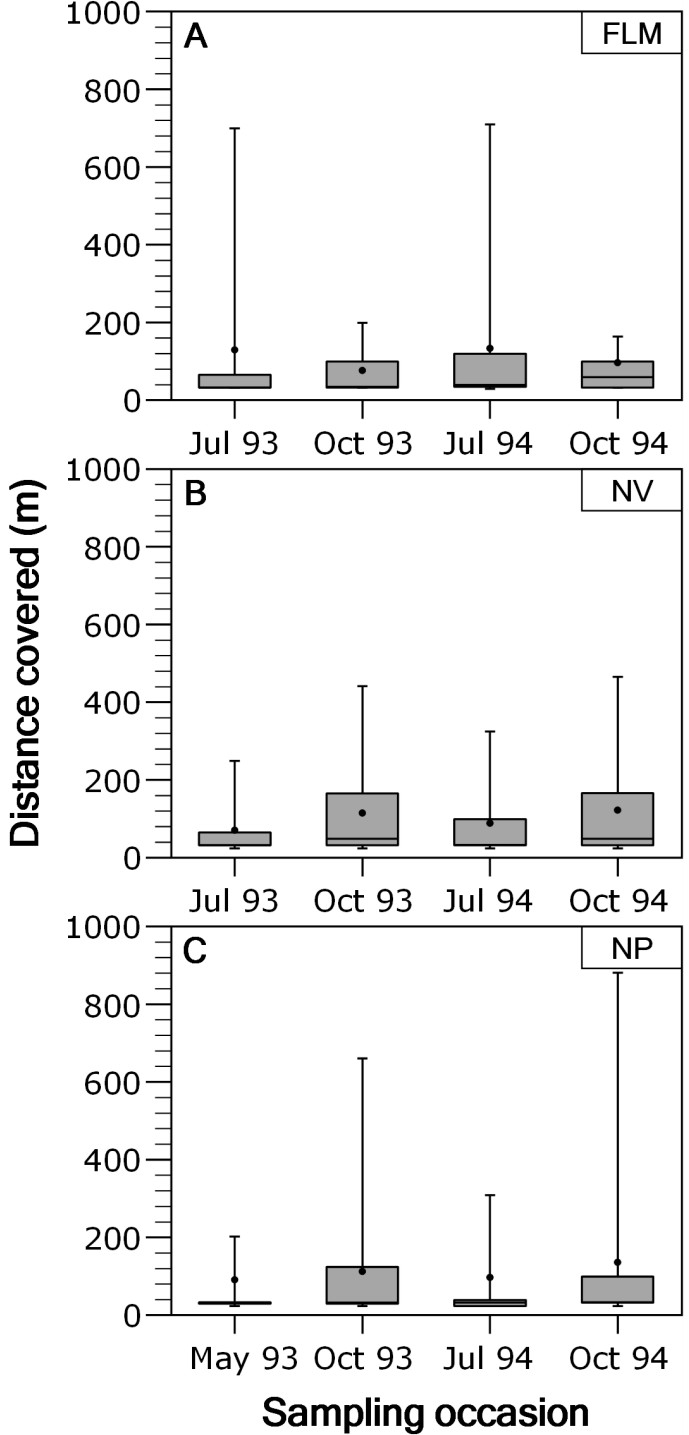

**Figure 3** **Box-plot of absolute distance moved, by brown trout individuals recaptured out from home section, per sampling occasion in the study streams.** The box corresponds to the 25th and 75th percentiles, the dark line inside the box represents the median, the error bars are the confidence intervals at the 95% confidence level, and the filled circle is the mean. (A) Flamisell. (B) Noguera Pallaresa. (C) Noguera Vallferrera.

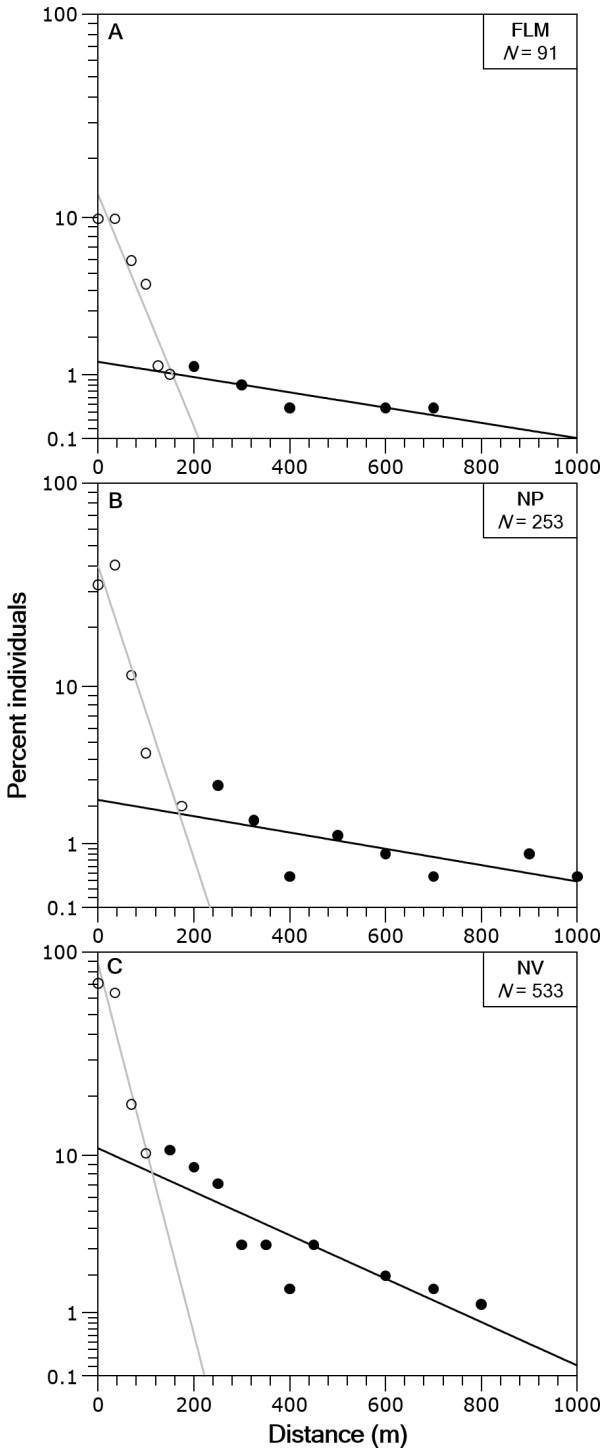

**Figure 4 Brown trout frequency distributions of the dispersal range fitted with a two-group exponential model in the study streams.** The solid lines are the linear regression fits for the estimated stationary (open circles) and mobile (filled circles) components of the fish populations. Distance classes are calculated from absolute values of distance moved, $y$-axis is in logarithmic scale. (A) Flamisell. (B) Noguera Pallaresa. (C) Noguera Vallferrera.

## Influence of biotic and abiotic factors on movements

Several biotic and abiotic factors were associated with brown trout movement patterns. Overall, the information theoretic analysis of the candidate set of GLMs selected eight plausible models (i.e., $\Delta AICc < 2$) to explain variability in departure ratio (i.e., proportion of individuals leaving section $i$ from the total number of recaptures of trout initially marked in section $i$) in relation to stream features. The best explanatory variables (SP values in Table 3), were water depth, fish cover and fish biomass. Interestingly, fish biomass was preferentially selected over density, despite the high correlation between them (Pearson's $r = 0.94$, $N = 246$, $P < 0.0001$). All three variables (water depth, cover and biomass) were negatively related to departure ratio; in contrast, organic substrate, water velocity, substrate coarseness and fish density had a positive relationship with departure ratio. Season had the weakest relationship with trout departure ratio. However, the lower correlation between observed and predicted values and larger parameters bias indicated some differences among streams (Table 3, Table S1). When analysed separately by streams, similar patterns were observed, with some variations linked to specific stream habitat characteristics (Table 1, Table 3). For instance, in the FLM stream where higher departure ratios (ANOVA, $F_{2,243} = 11.67$, $P < 0.0001$) and lower biomass and density values (MANOVA, Wilks's $\lambda = 0.513$; $F_{4,484} = 47.92$; $P < 0.0001$) were reported, physical features (i.e., fish cover, substrate coarseness and water depth) and fish biomass were the best explanatory variables. Seasonal variation in departure ratio confirmed previous observed results (Fig. 3); thus, while no seasonal differences were observed in the FLM stream, autumn departure ratio (i.e., pre-spawn movements) was higher in the NV stream, and lower in the NP stream. In the NV stream, along with fish biomass and season, the most important variable was water velocity, all negatively related to departure ratio. In the NP stream, fish density and biomass showed a contrasting pattern (Table 3), which might be explained by differences in fish length, since NP fish were the smallest (ANOVA, $F_{2,3070} = 184.60$, $P < 0.0001$).

Moving brown trout were significantly larger than non-moving individuals in the three study streams ($P < 0.029$ and $\eta^2 > 0.16$, in all cases), but distances moved were not significantly correlated with trout fork length ($P > 0.15$, in all cases). In both FLM and NV streams, specific growth rate of moving trout ($\mu \pm$ standard error: $0.132 \pm 0.006\%$ day$^{-1}$ in the FLM and $0.057 \pm 0.002\%$ day$^{-1}$ in the NV) was significantly higher than non-moving trout ($0.103 \pm 0.005$ and $0.050 \pm 0.001\%$ day$^{-1}$) (ANCOVA, $F_{1,242} = 5.73$, $P < 0.05$, $\eta^2 = 0.19$, in FLM, and $F_{1,1262} = 6.07$, $P < 0.05$, $\eta^2 = 0.28$ in NV), but no significant differences were observed in the NP stream ($0.048 \pm 0.002\%$ day$^{-1}$ for moving fish and $0.052 \pm 0.002\%$ day$^{-1}$ for non-moving fish) ($F_{1,327} = 1.06$, $P = 0.31$).

## DISCUSSION

### Movements and dispersal patterns

Brown trout exhibited limited mobility and few fish performed long-range movements. Between successive sampling events, a significant proportion of fish moved from their home section, but most of them moved less than 100 m and settled in contiguous sections (i.e., adjacent channel units). Individuals move in response to changing resource abundance and

Aparicio et al. (2018), *PeerJ*, DOI 10.7717/peerj.5730

**Table 3 Model averaged parameter estimates, by means of an information theoretic approach, from GLMs analysis of the influence of stream features of sampling sections on brown trout departure ratio (see text for definition).** FLM: Flamisell, NP: Noguera Pallaresa, and NV: Noguera Vallferrera. Model-averaged regression coefficients ($\beta$) are parameter coefficients averaged by model weight ($w_i$) across all candidate models ($\Delta AICc < 2$) in which the given parameter occurs; selection probability (SP) indicates the importance of an independent variable, and parameter bias is the difference between the averaged estimates ($\beta$) and the full model coefficients. The number ($N$) of candidate models ($\Delta AICc < 2$) and Pearson's correlation coefficient ($r$) between observed and model predicted values are also shown.

| Model parameter | All rivers model $N = 8; r = 0.47$ | | | FLM Model $N = 3; r = 0.79$ | | | NP Model $N = 12; r = 0.62$ | | | NV Model $N = 5; r = 0.72$ | | |
|---|---|---|---|---|---|---|---|---|---|---|---|---|
| | $\beta$ | SP | Bias | $\beta$ | SP | Bias | $\beta$ | SP | Bias | $\beta$ | SP | Bias |
| Intercept | 74.999 | | 0.237 | −73.138 | | −0.227 | 4.288 | | 4.772 | 91.815 | | 0.030 |
| Mean Depth (cm) | −1.442 | 1.000 | −0.021 | −2.745 | 1.000 | −0.239 | −0.123 | 0.208 | −2.289 | 2.352 | 0.224 | 0.019 |
| Fish Cover (%) | −1.109 | 1.000 | −3.069 | −1.772 | 1.000 | 0.242 | −1.339 | 1.000 | −0.396 | −0.264 | 0.366 | −0.239 |
| Fish Biomass (kg ha$^{-1}$) | −0.017 | 0.623 | −0.782 | −0.097 | 1.000 | −0.515 | −0.025 | 0.148 | −3.881 | −0.038 | 1.000 | −1.123 |
| Organic Substrate (%) | 0.118 | 0.182 | −5.555 | 1.246 | 0.126 | −2.124 | 0.390 | 0.292 | −2.289 | 0.761 | 0.464 | 0.104 |
| Water Velocity (m s$^{-1}$) | 0.991 | 0.153 | −10.204 | −5.704 | 0.323 | 0.867 | 5.981 | 0.318 | −0.923 | −55.562 | 1.000 | −0.214 |
| Fish Density (Ind ha$^{-1}$) | 0.012 | 0.147 | −29.545 | | *Not selected* | | 0.014 | 0.891 | −1.178 | | *Not selected* | |
| Substrate Coarseness | 0.305 | 0.104 | 6.702 | 46.146 | 1.000 | 0.021 | 2.951 | 0.339 | −1.425 | 4.977 | 0.212 | 0.004 |
| Pre Spawn Season | −0.125 | 0.077 | 2.550 | | *Not selected* | | −9.752 | 1.000 | 0.112 | 14.314 | 1.000 | −0.131 |

quality, thus movement extent should depend on the distance to a suitable habitat (*Fausch et al., 2002*), with longer movement distances presumably occurring when suitable habitats are widely spaced (*Wiens, 2001*). Therefore, the low mobility exhibited by brown trout from this study could indicate that the different habitats required to complete its life cycle are in spatial proximity (*Northcote, 1992*; *Gowan et al., 1994*; *Albanese, Angermeier & Dorai-Raj, 2004*). Spatial habitat heterogeneity is usually higher in small streams than in larger rivers, where habitat units are more widely spaced (*Gorman & Karr, 1978*), and habitat heterogeneity reduces territory size and mobility (*Heggenes et al., 2007*). Consequently, limited movements of no more than a few hundred meters are frequently reported for trout populations in small streams (*Heggenes, 1988*; *Knouft & Spotila, 2002*) when compared to those from large and long rivers, where longer movements, up to several kilometres, have been observed (*Clapp, Clark Jr & Diana, 1990*; *Young, 1994*; *Zimmer, Schreer & Power, 2010*). Therefore, the size of the study streams could have influenced the limited range of movements observed (*Woolnough, Downing & Newton, 2009*).

Although there was substantial intra-population variation in movement behaviour, the studied populations were dominated by stationary individuals, in concordance with most of the studies on movement in stream salmonids (*Rodríguez, 2002*, and references therein). Our results also match the estimated movement parameters (i.e., proportion of stationary and mobile individuals, and mean dispersal distances) provided by predictive models for stream-resident brown trout (*Radinger & Wolter, 2014*), for both the stationary and the mobile component. The displacement range of the stationary component was less than 50 m, and thus, is considered as a restricted movement (*Rodríguez, 2002*). Despite being in low proportion, mobile individuals are important since they are responsible for exchange between populations and successful colonization of new habitats (*Vera et al., 2010*; *Kennedy, Rosell & Hayes, 2012*).

Many studies dealing with brown trout movements showed a seasonal increase in mobility due to upstream spawning migration and downstream movement for overwintering (*Clapp, Clark Jr & Diana, 1990*; *Meyers, Thuemler & Kornely, 1992*; *Ovidio et al., 1998*). Our results showed temporal differences in the distance covered by moving fish, thus suggesting that seasonal changes in environmental conditions or the reproductive cycle of fish affected trout movements. There was an increase in distances moved in autumn surveys in both the NP and the NV stream, just before the spawning period, although this cannot be considered a true spawning migration because movement did not involve most of the population nor were distances moved long-range. In streams where spawning sites are located in or near the rearing habitats, as occurred in the studied reaches where gravel beds were widely distributed, spawning-related movements may be minimal or involve short distances (*Northcote, 1992*; *Nakamura, Maruyama & Watanabe, 2002*).

Movement data were based on recaptured (i.e., surviving) tagged fish, thus several potential sources of bias could have affected our results. For instance, VI-tag insertion and adipose fin clipping may result in different behaviour or mortality rate compared to the non-tagged fish. However, studies on salmonid species suggest that adipose fin clipping and VI tag insertion do not affect condition, growth or survival (e.g., *Bryan & Ney, 1994*; *Shepard et al., 1996*), thus the behaviour of tagged fish is representative of the population.

Another potential source of bias could be related to the relatively low percentage of VI-tags recovered, not only due to tag loss but also to natural mortality, which is expected in a nearly two years study. Although it was not measured, annual mortality for adult brown trout is estimated to be 43–72% in similar Pyrenean rivers (*Gouraud et al., 2000*). Tag loss may bias abundance estimates from mark-recapture methods (*Frenette & Bryant, 1996*), but movement estimates are not biased since differences in behaviour between tagged and tag-loss fish are not expected. Therefore, both mortality and tag loss only affect movement estimates by reducing total data gathered, but this was avoided by maximizing sample size increasing the number of fish tagged. Trout leaving the sampling area probably also accounted for a percentage of missed recaptures, thus leading to an underestimation of long-range movements (*Gowan et al., 1994*). However, the high proportion of recaptures of fin-clipped fish suggests that the emigration from the study area was proportionally low compared to the trout population monitored (*Gowan et al., 1994*).

## Relationship of habitat and biotic factors with brown trout movements

Fish movements are mediated by biotic and abiotic factors affecting individual fitness (*Gowan et al., 1994*; *Railsback et al., 1999*; *Bélanger & Rodríguez, 2002*). Our results showed that departure ratio was not consistently linked to particular habitat features, suggesting that movement behaviour probably does not depend on a single parameter but on a complex combination and availability of several factors. For instance, fish inhabiting shallower sections in the FLM stream showed higher departure ratios than those from deeper waters. Among the study streams, the FLM had the lowest mean water depth, therefore deeper waters, which confer greater protection against predators (*Lonzarich & Quinn, 1995*), were a valuable resource motivating trout movement. In the same sense, sections with higher fish cover in the FLM and NP stream showed lower departure ratios, probably because of the refugia provided from predators and fast currents (*Harvey, Nakamoto & White, 1999*; *Ayllón et al., 2014*). Finally, the negative effect of water velocity on departure ratio in the NV stream could be mediated by an increase in prey delivery rate in high velocity areas (*Leung, Rosenfeld & Bernhardt, 2009*), thus promoting residency.

Movements of stream fishes have been associated with fish size and growth rates. There is evidence from previous studies that movement distances increase with fish length, particularly for trout larger than 300–400 mm FL (*Meyers, Thuemler & Kornely, 1992*; *Young, 1994*; *Quinn & Kwak, 2011*). In the three study streams moving fish were larger than non-moving trout, but there was no relationship between distance moved and fish size. Results are probably skewed by the scarcity of large fish, since only few individuals (<0.5%) exceeded 300 mm FL, or the fact that large dominant fish could displace subordinate individuals, thus leading a movement cascade of fish of all sizes and obscuring length-related patterns (*Gowan & Fausch, 2002*; *Railsback & Harvey, 2002*). Mobility is energetically expensive and increases risk of predation since fish have to move through unfamiliar space (*Pépino, Rodríguez & Magnan, 2015*). Nevertheless, mobility appeared to confer an advantage on trout growth rate. Several authors have reported

that, when resources are scattered, mobile fish maximize food supply and compensate energetic costs of movement, thus increasing growth rate (e.g., *Hilderbrand & Kershner, 2004*; *Závorka et al., 2015*).

## Management implications

Most of the remnant brown trout populations of the Mediterranean lineage are confined to streams where many artificial in-stream barriers have been built for hydropower generation. For example, more than 400 weirs and dams are distributed in the Spanish Pyrenean rivers (*Espejo & García, 2010*), mainly for small-scale hydropower production. Barriers can restrict movement pathways among critical habitats for spawning, feeding or refuge (*Dunham, Vinyard & Rieman, 2011*), thus stream longitudinal connectivity is a key consideration in the management and conservation of the brown trout. The low rate of average movement at population level reported here suggests that management of brown trout populations in headwater streams, similar to those studied here, should focus on the conserving high quality habitats, whereas stream connectivity may not be as critical since relatively small stream length may provide sufficient habitat to meet all life-history needs. However, connectivity could become important when other anthropogenic impacts are present or the population verges the threshold of the minimum viable size (*Hilderbrand & Kershner, 2011*). Then, the importance of medium and long distance movements may be higher for population persistence, by allowing fish to withstand locally unfavourable conditions or favouring the recolonization after the disturbance (*Fausch et al., 2009*).

Finally, since brown trout is an important game species, in populations with limited dispersal fishing regulations could exert a high influence on trout abundance. Overfishing and depletion of fishery stocks seem more probable in stream reaches inhabited by trout populations with low mobility because substantial immigration of new individuals or eventual displacement of fish to safer areas (i.e., closed fishing reaches) are infrequent. To improve native trout abundance in streams with significant fishing pressure, no-kill regulations are advised or, at least, hardening existing regulations to avoid overexploitation.

# ACKNOWLEDGEMENTS

Authors wish to thank the many people who assisted to fieldwork, especially to F Casals, MA Puig, JM Olmo, MJ Vargas, J Malo, C Franco and J Laborda. We are grateful to the anonymous reviewers for their feedback which was very helpful in improving the manuscript.

## Funding

This research was supported by the Biodiversity Conservation Plan of ENDESA through the project PIE 121043-00-FECSA. The funders had no role in study design, data collection and analysis, decision to publish, or preparation of the manuscript.

## Grant Disclosures

The following grant information was disclosed by the authors:
ENDESA: PIE 121043-00-FECSA.

## Competing Interests

Rafel Rocaspana is an employee of Gesna Estudis Ambientals S.L. Carles Alcaraz is an employee of the IRTA (Institut de Recerca i Tecnologia Agroalimentàries).

## Author Contributions

- Enric Aparicio conceived and designed the experiments, performed the experiments, analyzed the data, contributed reagents/materials/analysis tools, prepared figures and/or tables, authored or reviewed drafts of the paper, approved the final draft.
- Rafel Rocaspana analyzed the data, contributed reagents/materials/analysis tools, authored or reviewed drafts of the paper, approved the final draft.
- Adolfo de Sostoa and Antoni Palau-Ibars conceived and designed the experiments, performed the experiments, authored or reviewed drafts of the paper, approved the final draft.
- Carles Alcaraz analyzed the data, contributed reagents/materials/analysis tools, prepared figures and/or tables, authored or reviewed drafts of the paper, approved the final draft.

## Animal Ethics

The following information was supplied relating to ethical approvals (i.e., approving body and any reference numbers):

The Departament de Medi Ambient i Habitatge de la Generalitat de Catalunya (current Departament d'Agricultura, Ramaderia, Pesca, Alimentació i Medi Natural) of the regional authorities of Catalonia provided authorization to conduct the research (SF/602).

## Data Availability

The raw data are provided as Supplemental Files.

## Supplemental Information

Supplemental information for this article can be found online at http://dx.doi.org/10.7717/peerj.5730#supplemental-information.

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
