# Peer review of "Movements and dispersal of brown trout (Salmo trutta Linnaeus, 1758) in Mediterranean streams: influence of habitat and biotic factors"

_PeerJ, doi:10.7717/peerj.5730_

## Round 0.1 · original submission · Major Revisions

This is an interesting analysis of a fairly extensive dataset on Brown Trout movements. There are a number of grammatical errors and awkward sentences, as pointed out by the reviewers. Please address all of the reviewer's comments.

Additionally, I think it would be valuable to discuss how the quite low tag retention rate might influence results, especially as it was size-dependent. You should also acknowledge that your models explain little variability in movement distances (r2 between ~0.2 and ~0.6) and that individual covariates may not be very powerful (many models with deltaAIC <2). Also discuss in more detail how the timing of spawning and possible spawning movements relate to sampling dates and the ability to identify actual spawning movements.

It would also be helpful to discuss the fact that you are just observing the movements of the survivors; it is possible of course that fish that were not observed again did not survive (or lost the tag). Consider devoting a paragraph to possible sources of bias in movement estimates: fish moved outside the study area (you cover this well in the current MS), tag loss, mortality, fish with different movement propensities may have different capture probabilities. Discuss whether these factors are likely to be size-dependent or movement propensity-dependent and how/whether they influenced your results.

Reviewer 1 ·

Basic reporting

There are a few awkward sentences and minor grammatical and spelling errors (some of these are mentioned in the "Comments for the author" below), but overall the writing is clear. The objectives are stated clearly, and the manuscript organization, including the format of figures, tables and raw data, is appropriate.

Experimental design

The authors tracked movements of many individually marked fished over ~2 years in three streams. The spatial coverage of study reaches in each stream was extensive. The data set is a strong point of the study.

Validity of the findings

The analytical methods seem appropriate given the aims of the study. The discussion seems generally sound but the last sub-section, on management issues, seems too speculative and could be shortened substantially.

Additional comments

Ttile:

Consider replacing the subtitle with: influence of habitat and biotic factors


Abstract:

"...and the influence of habitat and biotic factors of Mediterranean brown trout" influence on what? Do you mean "...on Mediterranean..."? (see also comments on l. 60 below).

"Mobile brown trout were significantly larger than non-mobile individuals and exhibited significantly higher growth rates in two of the streams." Please avoid using "significantly" this way; the term is meaningless without reference to effect and sample sizes.


l. 20: "Fish movement (like other animals) are a choice": unclear meaning and grammar.

l. 62-63: "When all habitats required to complete fish life-history cycle are in spatial proximity, lower movement distances are expected." Please provide a reference.

l. 66: which activities does the 18-24 month period refer to, exactly? (see comment on l. 99-101 below).

l. 67: how does "degree of mobility" differ from "extent of movements"?

l. 68: seasonal variations of which variable(s)?

l. 79: is the Segre a river or a stream? You use both terms, but the reported features suggest that "river" is more appropriate.

l. 99-101: the chronology of activities should be reported more precisely. "Samplings" seems vague: when, exactly, were marking and recapture conducted at each site? (e.g., you should have no recaptures in the first sampling session).

l. 209-210: Tag retention was low and you mention "fish mortality" (l. 206; was it quantified?), so it is unclear what you mean by "The large number of tagged fish prevented both tag loss and mortality effects."

l. 218-220: Please avoid mentioning "significance" without providing effect sizes; those P values are devoid of meaning without additional context, including the fact that your sample sizes (from the reported degrees of freedom for F) are in the hundreds. Large sample sizes often yield "statistically significant" effects of trivial magnitude.

l. 238-239: "river" can hardly be called an "explanatory" variable, because it does not generalize to other settings. Perhaps it is a covariate which you included to adjust for unmeasured variation among rivers?

l. 239-241: "Interestingly, fish biomass was preferentially selected over density, despite the high correlation between them (Pearson’s r =241 0.94, N = 246, P < 0.0001)" There is nothing unusual here; since the effects of the two variables are more or less equivalent, their partial coefficients can be very different than their marginal coefficients. It is probably not a good idea to include both variables in a single model. Also, high density and biomass are presumably strongly related to habitat suitability, but the models do not allow you to untangle these effects because biotic and abiotic effects covaried naturally in your design. You would likely have to experimentally control the two types of factor to avoid confounding their effects. These comments also apply to the statement on l. 255-257 ("Surprisingly,..."): the "contrasting patterns" are likely a simple consequence of your examining partial coefficients in strongly correlated variables.

l. 258-260: Again, statements based on P values, but no mention of effect sizes.

l. 261-264: what are the values reported in parentheses? Mean and s.e.? s.d? Also, for the NV, is the difference in SPG between mobile and less-mobile trout biologically interesting? The differences in reported means seem negligible both for the NV and the NP.

l. 272-273: It seems unwarranted to extrapolate to "Mediterranean streams" based on results from three neighbouring streams.

l. 314-316: "Our results showed that departure rate was not consistently linked to particular habitat features, but depended on stream-specific characteristics. Consequently, habitat features are important in influencing movement behaviour when present above or below a certain threshold. For instance, fish inhabiting shallower sections in the FLM stream showed higher departure rates that those from deeper waters." I do not understand these statements. First, there would seem to be a clear effect of habitat; second, where does the notion of threshold come from?; third, how do habitat features differ from "stream-specific characteristics"?

l. 333: "Mobility appeared to confer an advantage on trout growth rate." But which is the cause and which the effect?


Table 1:

Report s.d., percentiles, or some other appropriate measure of sample variation. The standard error of the mean is not useful as a habitat descriptor.

The units for biomass are incorrect.


Table 2:

Provide units for the lambda parameter.

·

Basic reporting

Lines 25-27: "Abundance" is not correct here. These are fewer fish of the same population that demonstrate longer movements. They are not a different group of fish.
Lines 62-64: The wording “complete fish life-history cycle” is awkward. Consider revising, perhaps: “..to complete the cycle of a fish’s life-history”.
Lines 64-65: errant use of a semicolon after “availability”.
Lines 65-76: Consider reordering this sentence to: “The objectives of this study were to employ mark-recapture approaches over a period of 18 – 24 months….”
Lines 76-78: delete “, it is the river”.
Lines 79-82: This text should be moved up to the 2nd sentence of the study area. Describe the Segre River first, then the tributaries.
Line 80: Is the 2,840 meters the elevation of the Pyranees or the change in elevation of the river? Unclear.
Figure 1. Caption needs to be improved to identify what the specific boxes are, and describe the abbreviations in the pictures.
Lines 95-98: How were the reaches and sections selected? Randomly? More information on the study design is needed.
Lines 99-101: Why the different sampling moths in 1993 in NP? How are these data compared with the other seasons? (July-October).
Lines 101-102: Were surveys done each year in autumn?
Lines 104-106: How was sex determined? Is the approach accurate-perhaps add a citation if available.
Lines 116-117: Consider breaking paragraph here.

Lines 118-119: Was substrate analyzed as a continuous variable? Was each category considered in the analysis separately? Weren’t some aspects correlated?
Line 120: “1985” needs parentheses around it.
Lines 145-146: The comma does not work appropriately here. Change text after “recorded” to :”..recorded, and only fish captured during at least three surveys () were included in the analyses”.
Lines 154-158: Should this be a separate objective? This is essentially a comparison of how brown trout movements in this basin compare to other populations of brown trout.
Lines 175-179: Here, it seems like a more elegant approach would be to use river as a random effect. Are there reasons this was not used? Also, in this analysis, it would be easier to interpret the beta estimates across attributes and streams if they were standardized (e.g, z-score
Line 193: Here, the authors do not use a comma for numbers greater than 999, elsewhere they do. Please correct based on the journal specifications.
Lines 193-197: How were the capture probabilities (with range) estimated? Were there from the multipass surveys?
Lines 203-205: Wouldn’t it be more accurate here to simply refer to these as variability in recruitment and probability of capture?
Lines 209-201: This last sentence is not clear.
Lines 218-221: Is this for all sized fish? It would be expected for larger brown trout making spawning migrations, but are smaller fish making these migrations? Given the lack of understanding of movement in this region, quantifying movement patterns by life-stage (juvenile vs. adult) would be interesting to help identify habitat and life-history expressions across stages—i.e., to provide insights into the needs for the species.
Line 234: The use of the term “influence” is not appropriate-consider using “associated”.
Lines 238-239: The use of the term “best” here is not appropriate given the number of candidate models supported by the data. Also, the authors should provide a table with (at a minimum) all plausible models (<2 delta AIC). This could be in the supplemental, but right now it is hard to understand what “best” means.
Lines 239-240: Was density included with models using biomass? It seems like either using a correlation analysis prior to the analysis or testing for initial support of one or the other via AICc, while including all other variables, would be the best approach. It is unclear if considered in the same model, which wouldn’t be appropriate given the correlation.
Line 241: it is unclear which “three variables” the authors are referring to.
Lines 245-247: Given that there was support for a stream effect, why do the authors discuss the “global” model that does not include a stream effect ( Lines 234-245)? A more streamlined approach and discussion in the methods would help here. For example, “we initially evaluated an effect of stream…”. If there is support, then all subsequent analyses should be performed with stream. Also, it is unclear why the authors do not consider size in this analysis? Or it would be appropriate to use size as a group-here, perhaps juvenile or adult?
Lines 258-260: How is this possible?
Lines 278-281: Here, the authors seem to ignore the fact that the stream with the highest range for sedentary and mobile fish (FLM) is also the shallowest and not the largest in width. While it does have slightly larger baseflows. How do these sizes actually compare with those cited from the literature?
Lines 299-303: Would the analyses differ here if the authors delineated their analysis by size?
Line 317: “that” after “rates” should be changed to than?
Lines 333-336: Mobility also has its costs in terms of predation, which is not discussed.
Lines 339-359: Given that the take home of this paper is that most fish don’t move, how is connectivity important? I am being devil’s advocate here, but how do your results provide compelling evidence for connectivity and other management activities. What about habitat restoration?
Table 3: The authors should change the wording from “best” in line 9. With so many models supported by the data, this terminology is not appropriate.
Figure 4 and corresponding analyses. In looking at the results here, it appears that a linear model was used, yet the authors claim that a 2-group exponential model was used. Are these lines from the exponential analyses?

Experimental design

Overall, the experimental design is appropriate.

Validity of the findings

Here, I suggest the authors consider 2 things: (1) providing methods to describe a logical procession of parameters to be included in the GLM analyses. E.g., test if there is a “stream effect”, then proceed with all subsequent analyses with stream effect included. Here, the authors, consider the results of the model that does not include stream, then report the results of the stream. The neat part of this analysis is that it demonstrates that habitat appears to influence movement patterns. (2) the authors should delineate the analyses by size. This does get tricky with considering size of fish marked in year a and recaptured in year a+1 because fish are bigger and older in year a + 1. However, really trying to inform management would consider how movement varies by life stages and what factors may be associated with movement by life stage.

Additional comments

In general, a well written manuscript that does have interest to a broad number of fisheries biologists. I do think the authors need to consider altering their approach in the statistical modeling (see specific comments) and comments above in “validity of findings”. I also suggest the authors should highlight the differences in movements across streams. The dispersals and movement metrics are considerably higher in the FLM stream. These results should demonstrate the importance of place and while generalizations may occur (e.g., most fish don't move), the different movements and importance of habitat are interesting. Finally, I think the authors could do a better job of explaining what these results mean in terms of management. The implications are that most fish don't move, yet the authors discuss the role of connectivity?

---

## Round 0.2 · Minor Revisions

Thank you for your careful attention to the reviewers' and my suggestions. While reviewer 1 is satisfied with the manuscript, reviewer 2 still has a number of concerns that need to be addressed. Please carefully consider and address all of reviewer 2's comments.

Reviewer 1 ·

Basic reporting

No comment

Experimental design

No comment

Validity of the findings

No comment

·

Basic reporting

No comment reported here, but see below.

Experimental design

Lines 186-191: Why did the authors consider the MANOVA analyses to quantify "variations in fish density and biomass among streams.."? This analyses, which appears to describe how biotic and abiotic factors affect these parameters are not in line with the objectives. How does this link directly to the movement data?

Validity of the findings

Lines 225-227: Why were these size classes used here and why different than in the previous results? The authors should align the information for consistency.

Lines 235-237: Does this analysis only include those fish that moved? Or are those that did not move (i.e., did not leave home section; resident) included? If the latter, than perhaps a zero-inflated regression may be appropriate with length as a covariate as ~54% did not move.

Lines 241-242: How specifically was kurtosis measured? This is not described in the methods.

Lines 243-250: These results are confusing due to the definitions presented early. For example: (1) In Lines 160 -161, the authors stated that “Trout were classified as mobile (individuals leaving the home section) and non-mobile (sedentary individuals not leave the home section individuals).”; and (2) In Lines 148-149, the authors stated “A movement distance of 0 m was assigned to individuals captured and recaptured in the same section”. Thus, sedentary individuals movement must = 0. Thus it is unclear how Figure 4 was created as the sedentary individuals do not move. This is also true for lines 246-248-how are these values calculated based on the definition of sedentary?
It seems this confusion is clearly based on the definitions presented in the methods. The authors should provide clear criteria and justifications for describing individuals as sedentary or mobile.

Additional comments

Overall, a great improvement from the last version. A couple of questions linger. First, there are a number of details that need to be clarified and confusion in how ‘sedentary’ is defined, then used in the analyses (see below). Next, despite the consideration of length and use of an ANCOVA, the authors should consider breaking their analyses into juveniles and adults based on known information of sexual maturity. This may provide more insight into the observed patterns. Specific comments are provided below.

Specific Comments
Lines 37-38: Consider rewriting to: "Fish movement can be highly variable...". The current writing is given a sense of being-as if 'movement' is exhibiting this variation.
Lines 53-56: consider breaking this sentence into two. The use of 'thus' is awkward. Another option would be to simply eliminate ", thus, while' and replacing this with 'with'.
Lines 60-61: 'focussed' is mis-spelled.

Also, here, what did Vatland and Caudron find?
Lines 64-65: 'habitats' should be singular.
Lines 66-68: Consider changing the text from '..additionally predict a reduction' to 'predict additional reductions' and change 'distribution' to 'distributional'.
Lines 73-74: It seems contradictory to use the term 'resident' here. Are none of these fish "migratory".
Lines 75-77: How does the fact that the streams are small and with channel unit habitat that are close together mean that there is high habitat heterogeneity?

Why not change the text here to: " The three study streams are small, have high gradient, and considerable habitat heterogeneity. "

Lines 77-81: Consider moving this text to after the objectives and rewriting such that these statements are linked with your expectations/hypotheses.

LInes 77-81: Consider moving this text to after the objectives and rewriting such that these statements are linked with your expectations/hypotheses.

Lines 157-158: How specifically were these evaluated? Using statistics? If so, the authors should report the alpha value.
Lines 161-162: Again, here and throughout 'samplings' should be singular. Here it should also be changed to 'sampling events' or 'marking events' or something similar.
Lines 168-173: By definition, isn't the λs = zero? Given the definition of a sedentary is an individual that does not leave its home section (Lines 160-161).
Lines 193-194: The authors should consider breaking this paragraph before ‘The effects of biotic..” as this is a completely different analysis.
Also here, did the authors run separate analyses for each stream? This is not clear.
Lines 217-219: Were these estimates of recapture probabilities from the depletion estimates? This should be clarified here. Also, the authors need to state in the methods that depletion estimates (I presume) were computed for different size classes in the methods.

Lines 230-232: The authors should report the samples sizes here. How many fish were recaptured and included in the movement analyses?
Line 231-232: Here the authors use ‘per cent’ (which should be percent), but lower the authors use ‘%’. The authors should seek PeerJ’s guide to authors and be consistent throughout.
Lines 235-237: Does this analysis only include those fish that moved? Or are those that did not move (i.e., did not leave home section; resident) included? If the latter, than perhaps a zero-inflated regression may be appropriate with length as a covariate as ~54% did not move.
Lines 237-240: consider changing text after ‘(Figure 3),’ to ‘..(Figure 3), but the seasonal differences were not significant in the FLM stream…’.

Lines 254-157: The term ‘departure rate’ insinuates a temporal component. The authors should consider a different term-preferably one that coincides with how others in the literature describe movement in this context.
Lines 258-260: Given the high correlation of the fish density and fish biomass, it would be inappropriate to include in the linear model (due to the assumptions of linear models). Were correlations evaluated prior to the analyses?
Lines 272-274: The text here “In the NV stream, together to fish biomass and season the most important variable was water velocity, both negatively related to departure rate.” is not clear. Doesn’t the MANOVA approach consider both fish biomass and density? Also, it would be helpful if the authors described the relationship in terms of actual biology where needed. For example, season is negatively related to departure rate? What does that mean? More in autumn?

In the same context, where years not included –i.e., seasons across years were grouped?

Lines 291-292: Change the text from “leave out” to either ‘move’ or something similar.
Lines 293-294: Consider changing “Movements of individuals are responses to..” to “Individuals often move in response to..”.
Lines 307-313: This text is ~redundant with the previous paragraph and should be combined with this previous text.
Lines 322-325: A word is missing between “because” and “did”.
Also, here, what is a “true spawning migration”? If the habitats near fishes provide all of the habitat to meet the fish’s life cycle (spawning to rearing to spawning) is migration necessary? This analysis would greatly benefit from splitting the movement analyses between likely juveniles and likely adults? Is this possible? This would provide more clarity to these results. The fact that length in the ANCOVA was not significant does not mean that juvenile and adult movements were not different. These are binary.
Table 2: There should be consistent use of significant digits in the columns. Dispersal range should be 1 significant digit, particularly given the scale of measurement (~between median section differences).
Table 3: There should be consistent use of capitalization in the table-‘Water velocity’ is no capitalized but all others are all capitalized. Also, the authors should consider using only 2 significant digits in the table as it is difficult to follow as is.

In addition, the use of the ‘bias’ analysis here is incorrect. If the ‘full model’ has very limited support, why is it contrasted with the model averages? The ‘bias’ information is not all that informative. The authors should instead consider a table showing the plausible models (with AICc and Wi), and another showing the model-averaged values with the beta values with SE and the importance values.

---

## Round 0.3 · Minor Revisions

Thank you for your careful consideration of comments from reviewer 2. Based on your responses, I chose not to send the MS out for further review and took another look myself. There are a number of small editorial changes that need to be made (e.g. many errors in the reference list), and a couple of small issues that need to be addressed.

I added comments to your track changes file and have asked PeerJ to send you the Word file directly so you can see comments (which don't show up in the attached PDF file) and easily make changes. Please address all of the comments and editorial suggestions in your resubmittal.

---

## Round 0.4 · accepted · Accept

Your article is now Accepted although I notice that there are still errors in the reference section. I highlighted them in the attached pdf.

Please work with the Production team to fix these (reference manager) errors and check the reference section very carefully before publication.

#